# Recent Updates and Advances in Winiwarter-Buerger Disease (Thromboangiitis Obliterans): Biomolecular Mechanisms, Diagnostics and Clinical Consequences

**DOI:** 10.3390/diagnostics11101736

**Published:** 2021-09-22

**Authors:** Bahare Fazeli, Daniela Ligi, Shayan Keramat, Rosanna Maniscalco, Hiva Sharebiani, Ferdinando Mannello

**Affiliations:** 1Immunology Research Center, Inflammation and Inflammatory Diseases Division, School of Medicine, Mashhad University of Medical Science, Mashhad 9177948564, Iran; fazelib@mums.ac.ir (B.F.); hivasharebiani@yahoo.com (H.S.); 2Vascular Independent Research and Education, European Organization, 20157 Milan, Italy; 3Unit of Clinical Biochemistry, Department of Biomolecular Sciences, Section of Biochemistry and Biotechnology, University “Carlo Bo” of Urbino, 61029 Urbino, Italy; daniela.ligi@uniurb.it (D.L.); r.maniscalco@campus.uniurb.it (R.M.); 4Hematology Department, Faculty of Medicine, Mashhad University of Medical Science, Mashhad 9177948564, Iran; shayan.sk1993@gmail.com

**Keywords:** Buerger’s disease, Thromboangiitis obliterans, oxidative stress, antioxidant capacity, blood count, lipid profile, cytokines, autoantibodies, hypercoagulation, infection

## Abstract

Thromboangiitis obliterans (TAO) or Buerger’s disease is a segmental inflammatory, thrombotic occlusive peripheral vascular disease with unknown aetiology that usually involves the medium and small-sized vessels of young male smokers. Due to its unknown aetiology and similarities with atherosclerosis and vasculitis, TAO diagnosis is still challenging. We aimed to review the status of biomolecular and laboratory para-clinical markers in TAO compared to atherosclerosis and vasculitis. We reported that, although some biomarkers might be common in TAO, atherosclerosis, and vasculitis, each disease occurs through a different pathway and, to our knowledge, there is no specific and definitive marker for differentiating TAO from atherosclerosis or vasculitis. Our review highlighted that pro-inflammatory and cell-mediated immunity cytokines, IL-33, HMGB1, neopterin, MMPs, ICAM1, complement components, fibrinogen, oxidative stress, NO levels, eNOS polymorphism, adrenalin and noradrenalin, lead, cadmium, and homocysteine are common markers. Nitric oxide, MPV, TLRs, MDA, ox-LDL, sST2, antioxidant system, autoantibodies, and type of infection are differential markers, whereas platelet and leukocyte count, haemoglobin, lipid profile, CRP, ESR, FBS, creatinine, d-dimer, hypercoagulation activity, as well as protein C and S are controversial markers. Finally, our study proposed diagnostic panels for laboratory differential diagnosis to be considered at first and in more advanced stages.

## 1. Introduction

Thromboangiitis obliterans (TAO) or Buerger’s disease is a segmental inflammatory, thrombotic occlusive peripheral vascular disease with unknown aetiology that usually involves the medium and small-sized vessels of young male smokers [1]. TAO has geographical distribution, and it is more common in the Middle East, Far East, South-East Asia, Eastern Europe, and South America [2]. However, there is still no explanation for the geographic distribution of TAO.

Due to the unknown aetiology of TAO, its diagnosis is still challenging. Pathology of the acute lesions is pathognomonic for TAO diagnosis [3]. However, the pathology study is possible on amputees. Even biopsies of superficial thrombophlebitis are not recommended in TAO patients due to the poor circulation of the limb and the risk of developing chronic ulcers at the site of biopsy.

Moreover, until recently, TAO could neither be suggested nor be diagnosed according to biological markers.

Therefore, several diagnostic criteria have been suggested for TAO diagnosis. In particular, one of the most acceptable in the countries where TAO is common is Shionoya’s clinical criteria [4]. These criteria refer to the clinical manifestation of the disease and the absence of atherosclerotic risk factors except for smoking (e.g., hyperlipidemia and diabetes). Since TAO is usually registered in low socioeconomic classes or countries with social, economic, or political crises [5], Shionoya’s clinical criteria are eligible for these patients to escape the considerable costs of imaging. However, Shionoya suggested these criteria in 1988, whilst the cut-off points for the definition of hyperlipidemia and diabetes in the 1980s were different from their current cut off points. Several biomarkers of atherosclerosis have been introduced and studied during the last decades, and some of them, like hyper-homocysteinemia, have been reported in both atherosclerosis and TAO [6,7].

However, some angiologists prefer to diagnose TAO by excluding other types of vasculitis and hypercoagulable states based on laboratory para-clinical findings alongside the imaging [8].

In this paper, we will review the status of studied biomolecular and laboratory para-clinical markers in TAO compared to atherosclerosis, vasculitis, and hypercoagulopathies.

## 2. Complete Blood Count (CBC)

Several studies reported anaemia, most commonly normochromic and normocytic type, in about 40% of TAO patients, particularly during the acute exacerbation of the disease [9,10]. Notably, a significant inverse correlation between haemoglobin (Hb) level and duration of smoking has been reported in TAO patients, which is not anticipated because a high Hb level would be expected in long-term, heavy smokers [9]. Moreover, one of these studies demonstrated that Hb from TAO patients had decreased over a short period of time, with a trend that did not resemble trends found in chronic anaemia, as observed in other types of vasculitis [9].

In another study, although the indirect Coombs test was positive in 70% of the TAO patients with high levels of LDH (Lactate dehydrogenase) and AST (Aspartate aminotransferase) and normal levels of ALT (Alanine aminotransferase), the definitive diagnosis of hemolytic anaemia was not confirmed, because of the normal level of bilirubin as well as the normal or high level of haptoglobin [10].

Although normal platelet count has been reported in TAO patients, thrombocytosis and thrombocytopenia have been reported in 6.4% and 2.9% of the patients, respectively [9]. Notably, thrombocytosis has been observed in those patients who underwent major amputation. Leukocytosis and neutrophilia have also been reported in TAO patients during the acute exacerbation of the disease [9].

Notably, leukocytosis, thrombocytosis, and normochromic and normocytic anaemia are usually seen in systemic vasculitis [11,12]. Moreover, leukopenia and thrombocytopenia are more common in vasculitis, whilst anaemia is also observed in systemic vasculitis.

Increased platelet response to serotonin has been described in TAO patients compared to healthy male controls. About half of the controls in this study were smokers [13].

In contrast to atherosclerosis or vasculitis, low mean platelet volume (MPV) has been observed in TAO patients [14]. MPV is an indicator of platelet activation, and there is strong evidence indicating that MPV is an important variable and that larger platelets have a higher thrombogenicity. It is implicated that the severe inflammatory condition has been associated with low MPV, which increases with anti-inflammatory therapy [14]. Interestingly, it has been reported that other metabolic disorders, such as high cholesterol and diabetes, have an association with increased MPV, and this could indicate a link between increased MPV and atherosclerosis [15].

The disease activity and severity in some vasculitis, such as Behcet’s disease, has been associated with higher MPV [16], and in some other vasculitis, such as Systemic Lupus Erythematous (SLE), has been associated with lower MPV [17].

A summary of the main findings is reported in Table 1.

## 3. Biochemical Markers

### 3.1. Lipid Profile

For years, TAO had been misunderstood as a kind of presenile atherosclerosis obliterans [18,19]. However, after the recognition of TAO as an individual disease, excluding the risk factors of atherosclerosis (e.g., diabetes and hyperlipidemia) became a part of the diagnostic criteria of TAO [4].

Notably, it seems that long term smoking influences the lipid profile, with higher cholesterol, higher low-density lipoprotein (LDL), and lower high-density lipoprotein (HDL) [20].

The serum cholesterol level in TAO patients has been reported ranging from 157 mg/dL to 225 mg/dL, whereas LDL and HDL levels were 94–112 mg/dL and 34–54 mg/dL, respectively [21,22,23].

In 2013, Hus et al. reported significantly higher cholesterol serum levels in cigarette-smoking TAO patients compared with non-smoking ones. However, regarding the serum level of cholesterol in these patients, it has not been reported whether it was in the normal range [21].

It is still unknown whether a low level of total cholesterol in TAO patients is due to any impairment in mitochondria function or because of the bias of excluding hyperlipidemic patients from TAO diagnosis.

### 3.2. Blood Sugar

During the 1930s, it was noticed that patients with TAO diagnosis did not have diabetes [24,25]. Nowadays, hyperlipidemia and diabetes are also known as excluding factors for a TAO diagnosis [4]. Even if a patient with a TAO diagnosis develops diabetes years after the initial diagnosis of TAO, the TAO diagnosis would be ruled out retroactively. In 1992, Papa et al. suggested not to rule out the initial diagnosis of TAO according to typical clinical manifestation and angiography if the patient develops diabetes years after TAO diagnosis [26].

In addition, a few cases of diabetes in patients with TAO diagnosis, confirmed by pathology without any atherosclerotic plaque, have been reported. Notably, the patients’ fast blood sugar (FBS) was not more than 113 mg/dL [21,27,28].

According to pathological studies of below-knee amputees of two diabetic cases with a diagnosis of PAD (peripheral artery disease), it has been reported that the diagnosis was TAO because of inflammatory thrombosis and the absence of atherosclerotic plaques. However, they had excluded those cases from the TAO group because of their being diabetic [22,29].

### 3.3. Oxidative Stress

Several studies indicate that oxidative stress promotes endothelial cell dysfunction and can directly influence vascular tone by decreasing nitric oxide (NO) bioavailability [30]. This condition activates a vicious circle because a malfunctioned vascular endothelium is a source of oxidative stress by itself. Moreover, besides vascular endothelium, it has been demonstrated that vascular adventitia is a major producer of oxidative stress, particularly in response to hypoxic stresses, where it has a key role in regulating vascular tone by targeting NO bioavailability [31]. It seems that the known risk factors of cardiovascular disease, including smoking, diabetes, and hyperlipidemia, lead to higher oxidative stress [32,33].

Notably, oxidative stress seems significantly higher in TAO patients than smoking habit-matched controls [34]. This means that other factors besides smoking are responsible for high oxidative stress in TAO patients. A low serum level of NO metabolites in TAO has been reported compared to smoking controls, which might be one reason for high oxidative stress in TAO. Nevertheless, it is not well defined if the low NO level is responsible for high oxidative stress in TAO or secondary to high oxidative stress [35]. Moreover, it has been reported that polymorphisms in the promoter region of nitric oxide synthase correlate with major amputation in TAO patients [36]. Interestingly, besides NO alterations in TAO patients, it has been demonstrated that the level of heme-oxygenase 1 (HMOX1), which is implicated in maintaining the backup of nitric oxide, is intact and significantly higher in TAO patients compared to controls [37].

Notably, a higher level of nitric oxide has been demonstrated in patients with coronary artery disease [38]. Moreover, increased NO production has been reported in some autoimmune conditions, including rheumatoid arthritis, systemic lupus erythematosus, and Behcet’s disease [39,40,41,42].

oxLDL, malondialdehyde (MDA), carbonyl protein (PC), superoxide dismutase (SOD), glutathione reductase (GR), myeloperoxidase (MPO), and coenzyme Q10 (co-Q10) are also important markers that play a role in oxidative stress status [43,44,45]. Interestingly, in 2010, Arslan et al. found that oxLDL was higher in patients with TAO than atherosclerosis [46].

Recently, a significantly lower level of MDA and SOD has been reported in TAO patients compared to smoker controls [35]. Notably, a higher level of MDA and SOD has been reported in atherosclerosis [33]. A higher level of MDA in SLE without any significant association with the disease activity has also been reported [47]. However, SOD activity has been reported as normal to significantly lower in SLE patients compared to controls [47,48].

No significant difference between PC, GR, and MPO levels has been demonstrated in TAO vs. controls [35]. However, a higher level of PC has been reported in both atherosclerosis and SLE [49,50].

Moreover, a higher level of glutathione reductase in atherosclerosis has been demonstrated, whilst no significant difference in GR level has been reported in SLE [51,52].

Further, a higher level of MPO in atherosclerosis and the active phase of several vasculitis, including Behcet’s disease has been reported [53,54].

Interestingly, the serum level of CoQ10 has been reported to be significantly higher in TAO compared to controls. The high levels of CoQ10 in TAO patients might be responsible for a lower level of MDA by increasing its catabolism [35]. Lower levels of CoQ10 have been noticed in atherosclerosis and in some autoimmune diseases [55,56].

### 3.4. Creatinine

Elevated serum creatinine concentration was proposed to be a marker for increased risk of cardiovascular disease mortality [57]. There are a few case reports about the involvement of renal arteries in TAO and consequently higher creatinine level [58]. Serum creatinine level in systemic vasculitis could also be high according to the involvement of the kidneys [59].

### 3.5. Catecholamine Concentration

Few studies have been performed to evaluate the level of catecholamines in TAO patients, either in group studies or in case reports. They showed that the level of catecholamines in patients with TAO increases. Of course, the effect of smoking on increasing levels of catecholamines has also been clearly shown, but Roncon-Albuquerque et al. demonstrated that in patients with TAO, cigarette smoking had different effects in plasma catecholamines compared to controls [27,60]. According to their results, smoking decreased noradrenalin in controls but increased in TAO patients. Although in TAO patients with sympathectomy smoking did not significantly change noradrenalin, there was a pronounced fall in adrenalin [60].

Compared with TAO, increased levels of catecholamines have also been reported in patients with atherosclerosis. Bauch et al. have shown significantly high plasma concentrations of epinephrine and norepinephrine in patients with coronary artery disease (CAD), and they considered these results as a hypothesis revealing that catecholamines may play a role in the development and subsequent complications of atherosclerosis [61]. In vasculitis, the association of catecholamines with the disease has been clearly investigated, and the results have led to the development of a group of vasculitis called Catecholamine-induced vasculitis. Catecholamine-induced vasculitis is well known. Sarathi et al. [62], studying the relationship between catecholamines and vasculitis, found a correlation between pheochromocytoma and aortoarteritis, and thus demonstrated the essential role of catecholamines in the development of aortoarteritis. This aspect has been recently confirmed by Toutai et al., who stated that high catecholamine levels may have a role in the etiopathogenesis of aortoarteritis in these patients [63].

### 3.6. Heavy Metals

Food, smoking, and air pollution are the major sources of lead, cadmium, and other toxic heavy metals such as arsenic [64]. Although the cardiovascular consequences of metal toxicity have not been published widely enough, it has been demonstrated that heavy metals induce vascular dysfunction by inducing high oxidative stress, reducing endothelial nitric oxide synthase activities, and enhancing the phosphorylation of myosin light chain kinases [65].

Various studies indicated that heavy metals in tobacco and other sources might be one of the causative factors for TAO development. This is particularly true for cadmium, lead, and arsenic [66].

An association between lead and cardiovascular disease has been recognized and considered an established risk factor for oxidative stress, inflammation, and the triggering event of atherosclerosis [67].

Moreover, the role of cadmium as a cardiovascular disease risk factor, especially for coronary disease, has been indicated [68]. In addition, the role of heavy metals in the formation of vasculitis has been reported [69]. Albert et al., studying Wegener’s granulomatosis (WG), found that mercury and lead have been associated with the etiopathogenesis of WG [70]. On the other hand, it has been discussed that lead and its possible pro-inflammatory action, including the influence on pro-inflammatory cytokine production and its effective role in causing vasculitis, can be considered [71].

### 3.7. Bilirubin

Generally, abnormal bilirubin levels are considered a sign of hepatic disorders. Although bilirubin has long been believed to be an excretory metabolite, one of the most critical studies on the association between bilirubin and cardiovascular system pointed to the effective role of bilirubin in preventing atherosclerosis [72]. In 1994, Schwertner et al. found that lower bilirubin levels were significantly associated with an increased risk of coronary artery disease [73]. On the other hand, it was observed that increasing bilirubin levels in cardiovascular patients are associated with a decreased risk of atherosclerosis [72,73,74]. In TAO, bilirubin has been reported in normal range [10]. In vasculitis, especially autoimmune vasculitis, the bilirubin can increase under the influence of autoimmune hemolytic anaemia (AIHA) [75,76].

Tovoli et al. also observed the increase of bilirubin due to autoimmune liver disorders (AILD) in small-vessel vasculitis [77]. Nevertheless, bilirubin plays an essential role in other vasculitis as a diagnostic marker, such as Behcet’s disease, where a decrease in bilirubin can be observed, according to Koca et al. [78].

### 3.8. Liver Function Test

The evaluation of liver function tests (LFT) provides many indications and is mainly performed in systemic diseases as a prognostic indicator. Notably, the patient’s age can influence the results of AST as a positive correlation and ALT as a negative correlation [79].

Higher levels of AST and normal levels of ALT have been reported in young TAO patients [75].

However, a higher level of ALT, as a marker of fatty liver and increasing the risk of insulin resistance, seems a risk factor for atherosclerosis [80].

Moreover, higher levels of liver enzymes have been noticed in several autoimmunities [77].

A summary of the main findings is reported in Table 2.

## 4. Inflammatory Biomarkers

### 4.1. C-Reactive Protein and Erythrocyte Sedimentation Rate

C-reactive protein (CRP) and erythrocyte sedimentation rate (ESR) have been usually observed in normal range in TAO patients [30]. Moreover, in some TAO diagnostic criteria, positive ESR and CRP have been considered factors for ruling out TAO diagnosis [81]. However, several studies reported high levels of CRP in TAO patients [75,82]. There is no study on ESR levels in TAO patients.

### 4.2. Cytokines

Although TAO is not known as systemic vasculitis, the serum levels of different cytokines have been evaluated in TAO patients. Significantly higher serum levels of pro-inflammatory cytokines, including TNF-alpha (TNF-α), interleukin-1 beta (IL-1β), and interleukin-6 (IL-6), have been reported in TAO patients compared to controls [83,84]. Nevertheless, in the work of Joras et al., the serum level of TNF-α had no significant difference between TAO and controls [85]. Moreover, cytokines of cellular immunity, including interferon-gamma (IFN-γ) and interleukin 12 (IL-12), cytokines of humoral immunity including interleukin 4 (IL-4), interleukin 5 (IL-5), and interleukin 13 (IL-13), have been reported in TAO patients compared to smoker controls [83].

Moreover, high serum levels of IL-17, -22, -23, and -33 and low IL-10 have been reported in TAO compared to smoker controls [14,86,87].

The levels of cytokine released and the T-cell mediated immunity in TAO could also be the result of different HLA patterns of expression in TAO patients, as reported by Shapouri-Moghaddam et al. [88].

### 4.3. Complement Component

High levels of C4 have been reported in patients with a severe form of TAO [89]. Notably, a high serum level of C3 and C4 has also been reported in atherosclerosis [89]. However, serum levels of C3 and C4 range from low to high according to the type of vasculitis [90,91].

### 4.4. Other Inflammatory Mediators

As a biomarker of cellular immunity, Neopterin has been reported to be significantly higher in the active phase of TAO compared to its quiescent phase and controls [92].

Toll-like receptors (TLRs) are the main components of innate immunity and have an essential role in the pathogenesis of autoimmune and autoinflammatory diseases. Notably, TLR4 has been reported to be significantly lower in the quiescent phase of TAO compared to controls. In patients with active TAO, no difference between the serum level of TLR4 in patients and controls has been found. No changes in serum levels of TLR2 in active and quiescent phase of TAO or between TAO patients and controls have been found [92].

High mobility group box 1 protein (HMGB1), which acts as a pro-inflammatory cytokine, has been reported to be significantly higher in TAO patients than healthy non-smokers [93]. Moreover, significantly increased intercellular adhesion molecule -1 (ICAM-1) and vascular cell adhesion molecule -1 (VCAM-1), key regulators of vascular permeability, have been demonstrated in TAO patients compared to healthy non-smokers [94,95].

Notably, a significantly high level of matrix metalloproteinase 9 (MMP-9) as a regulator of vascular permeability in TAO patients compared to non-smokers has been reported [93]. Similarly, high levels of different classes of MMPs have been reported in atherosclerosis and vasculitis [96,97,98].

A summary of the main findings is reported in Table 3.

## 5. Autoantibodies

Until recently, TAO is not known as a systemic vasculitis, and it is still unknown whether it is an autoimmune disorder [86,87]. However, the existence of several autoantibodies in some TAO patients has been reported, including anti-endothelial cell antibody (AECA), anti-elastin, and anti-collagen I and III antibodies [99,100,101,102]. In 1998, the presence of antineutrophil cytoplasmic antibodies (ANCA) was reported in patients with TAO [103]. The study found a significantly higher ANCA level in patients with severe clinical manifestations compared to patients with mild disease and controls [104]. However, several further studies have ruled out the presence of ANCA in TAO patients and their association with the disease [105].

In addition, high levels of anti-cardiolipin antibody and anti-Beta2 glycoprotein antibody have been reported in TAO patients [106]. Due to the fluctuation in the levels of these antibodies, the diagnosis of anti-phospholipid syndrome could not be established. However, it seems that TAO patients with high anti-cardiolipin titers tend to be younger and to suffer from a significantly higher rate of major amputations [106].

As reported in Table 4, some autoantibodies have also been described in vasculitis and atherosclerosis [107,108,109,110].

A summary of the main findings is reported in Table 4.

## 6. Thrombogenicity

### Hypercoagulation

Although hypercoagulability has been considered an exclusive factor for TAO diagnosis, several studies have reported thrombogenic risk factors in TAO patients.

For instance, lower clot permeability and prolonged clot lysis time in TAO patients compared to other types of peripheral arterial diseases (PADs) and smoker controls have been reported [111].

Also, increased thrombin formation has been reported in TAO patients [112]. Although a significantly higher level of urokinase-plasminogen activator (uPA) and lower level of plasminogen activator inhibitor I (PAI-1) have been reported in TAO, fibrinolysis seems to be impaired [21,113]. However, another study reported that PAI-1 was expressed along the internal elastic lamina, whereas urokinase-type plasminogen activator and MMP-3 were slightly expressed in intima and media [114].

Interestingly, it seems that D-dimer released from clots is markedly slower in TAO patients compared to other patients with PADs [111,115].

A significantly higher level of fibrinogen in TAO patients compared to patients with other types of PADs or healthy smokers has been reported [111,113].

Protein C and S deficiency has been considered as exclusive factors for TAO diagnosis [115]. However, two separate case reports have implied the co-existence of Protein C and S deficiency with TAO [116]. Moreover, a mutation in factor Leiden V has been reported in TAO patients via a few case reports. Interestingly, almost all of these case reports have thrombosis in large vessels or deep venous thrombosis, but the histopathology of the amputees was suggestive for TAO [117,118].

Hyperhomocysteinemia has also been reported in TAO patients [119]. Hyperhomocysteinemia can induce thrombogenicity by increasing vascular adhesion molecules and tissue factors, inhibiting fibrinolysis as well as inducing nitric oxide bioavailability impairment and platelet activation [120]. Nevertheless, its level in TAO patients compared to healthy smokers is controversial in different studies. Some studies have considered hyperhomocysteinemia as an independent risk factor for TAO development. However, in other studies, due to the absence of significant differences between homocysteine levels in TAO patients and healthy smokers, hyperhomocysteinemia as a consequence of smoking in TAO patients was not considered an individual risk factor [120,121]. On the contrary, in atherosclerosis, it has been described a normal coagulation profile [122].

On the other hand, in vasculitis, a hypercoagulable profile has been reported in both Behçet syndrome and AAV, in which the main cause is referred to NETosis and endothelial dysfunction, respectively [123]. Furthermore, high levels of d-dimer have been often reported in AAV, SLE, granulomatosis with polyangiitis (GPA), Henoch-Schönlein purpura, and Kawasaki disease [124].

Finally, increased levels of homocysteine have also been reported in atherosclerosis and vasculitis [125,126].

A summary of the main findings is reported in Table 5.

## 7. Infection

The footprint of infectious pathogens in TAO development was suggested early after the disease description by Leo Buerger [127,128]. However, until recently, two pathogens attracted the attention: oral bacteria and rickettsial infection. Although the high immunoglobulin titer against oral bacteria and rickettsia has been reported, it is not evaluated in the routine practice for TAO diagnosis [129,130].

A summary of the main findings herein described is reported in Table 6.

## 8. Genetic Background

Several studies have been conducted on the genetic background of TAO. Although susceptibility to TAO has been suggested to be at least in part controlled by genes involved in innate and adaptive immunity, vascular physiology, platelet function or coagulation pathways, the presence of any particular gene polymorphism is controversial.

One of the most critical limitations of studies on genetic background of TAO is the lack of sufficient sample size due to the rarity of the disease. Obviously, allele frequencies based on small sample sizes will increase the false-positive evidence for linkage [131]. Besides, there is a lack of supporting studies regarding any polymorphism claim for TAO. These limitations make the results of genetic studies on TAO not quite reliable. For instance, several studies have explored HLA typing in TAO since the 1990s [88,132,133,134,135,136]. Although few polymorphisms have been detected to be associated with TAO in each study, the results are quite regional. Besides HLA-DRB1*15, which is a common polymorphism in TAO patients from Iran and India [88,135], and HLA-DRB1*04, which is a common polymorphism in the TAO patients from Japan and Iran [88,136], the rest of the polymorphisms of HLAs in TAO patients have been varied in different regions. Notably, HLA-DRB1*15 has also been reported as a risk factor for PR3-ANCA associated vasculitis in African Americans [137], and HLA-DRB1*04 has been associated with several vasculitis [138,139,140]. No association between HLA-DRB1*15 and atherosclerosis has been reported. However, early cardiovascular events in smokers with HLA-DRB1*04 have been noticed [141].

Besides HLA typing, several studies on the single nucleotide polymorphism of nitric oxide synthase, endothelin 1, plasminogen activator inhibitor, platelet receptors, homocysteine, Factor V, prothrombin, and Toll-like receptor 4 and CD14 have been conducted on TAO patients.

However, until recently, among all studied polymorphisms, only two studies had reported the association of eNOS T-786 C polymorphism of endothelial nitric oxide synthase with TAO [36,142]. Interestingly, according to a meta-analysis conducted in 2015, there was no association between eNOS T-786 C polymorphism and vasculitis [143]. However, the association between eNOS T-786C and atherosclerosis seems to be controversial [144,145].

## 9. Discussion

Until recently, the aetiology and even TAO classification as a kind of PAD or small and medium-sized vasculitis has remained challenging.

Notably, during the 1960s, after reporting TAO cases with visceral involvement as well as evidence of atherosclerotic lesions in visceral vessels from the autopsy of some TAO cases, TAO was considered as a type of atherosclerosis obliterans (ASO) with slow collateralization of the lesions and also the retrograde extension of the lesions from distal to proximal in comparison with usual cases of ASO. However, later, according to several studies on large series of TAO patients, the clinicopathological findings and angiography manifestation were considered characteristic for TAO as a separate disease from ASO or vasculitis [1].

Maybe, when we consider a young smoker with thrombophlebitis migrans and vascular involvement of infrapopliteal arteries and upper limbs, with normal blood sugar, normal lipid profile, and normal blood pressure, with typical skip lesions and corkscrew collaterals in angiography, the TAO diagnosis seems to be easy. However, there are several cases where the disease manifestation might be after 45 years, the patient might not be smoker, have dyslipidemia or glucose intolerance, or even high blood pressure without any atherosclerotic plaque, but at the same time present without obvious corkscrew collaterals [27,28]. Hence, the diagnosis of such cases as TAO would not be easy.

Up to now, TAO diagnosis is based on ruling out ASO and other types of vasculitis [4]. Usually, such investigations are not pathological studies only because tissue sampling on ischemic limbs are not practical. Therefore, biomarkers are usually investigated for ruling out ASO and other types of vasculitis.

However, inflammation, thrombosis, infection, and the presence of some autoantibodies are a part of the pathophysiology of TAO, ASO, and vasculitis and could be detected in patients with any of these diagnoses. Hence, for challenging cases like TAO clinical manifestation in older patients or non-smoker patients, these biomarkers might not be helpful but also making the diagnosis more complicated.

Although some biomarkers might be common in ASO, TAO, and vasculitis, each disease occurs through a different pathway with different origins. On the other hand, according to our current knowledge, there is no specific and definitive marker for differentiating TAO from atherosclerosis or vasculitis [9]. Hence, the mechanistic investigation of biomarkers is more useful and applicable.

Mechanistic investigations are a kind of fuzzy logic that demonstrate the concept of partial truth, where the truth value may range between completely true (one) and completely false (zero).

Based on this approach, there are three groups of markers: (1) similar markers, (2) controversial markers, (3) and differential markers.

### 9.1. Similar Biomarkers

Pro-inflammatory and cell-mediated immunity (CMI) cytokines, IL-33, HMGB1, neopterin, MMPs, ICAM1, complement components, fibrinogen, oxidative stress, NO levels, eNOS polymorphism, adrenalin and noradrenalin, lead, cadmium, and homocysteine are the common markers.

Thrombosis, inflammation, presence of autoantibodies, endothelial injury or dysfunction, vasoconstriction, high oxidative stress, and infection are common pathological pathways in these diseases. Similar markers are derived from these common pathways. For example, NO level is associated with vasoconstriction [146], pro-inflammatory and cell-mediated immunity (CMI) cytokines, HMGB1, IL-33, neopterin, ICAM1, and complement components are most related to inflammation and infection [147,148,149]. Lead, cadmium, oxidative stress, and MMPs are related to cell damage [64,93]. Fibrinogen and homocysteine can be related to thrombosis [111,119].

### 9.2. Controversial Biomarkers

The controversial markers are platelet and leukocyte count, haemoglobin, lipid profile, CRP, ESR, FBS, creatinine, d-dimer, hypercoagulation activity, protein C, and S. Many factors including family history and genetics, underlying medical conditions, environmental conditions, consumption of some drugs, and accompanying other diseases have effects on the levels of these markers. Therefore, these factors should be considered along with these conditions and the complete history of patients.

For example, although most TAO patients have normal FBS and do not have diabetes, they can still have these two conditions simultaneously. It has been reported that some clinicians rule out the several years’ diagnosis of TAO because of high levels of FBS that recently occurred [26]. Even if the mechanism of disease decreases blood sugar and lipids, the basic levels of these molecules are different in each patient, depending on their genetic and history. Therefore, we see different levels of FBS and lipid profile in these patients [20,21,22,23,27,28]. These differences should not cause the early rule out of TAO diagnosis. In these cases, we should consider the whole data of each patient, including their signs and symptoms, pathology results, angiography, and other biomarker test results. For example, in the case of lipid profile, evaluation of TLR4 level, ox-LDL, and MDA can be helpful for the differentiation of TAO and atherosclerosis [35,92,93].

Infection, some drugs, and underlying conditions can have effects on platelet and leukocyte count [150]. Therefore, all of these conditions should be considered case by case, personally.

Anaemia is one of the most controversial conditions in these diseases. Smoking can induce high levels of haemoglobin in smokers. However, in TAO patients, several cases of anaemia have been reported, and notably, the haemoglobin level is associated with the prognosis of TAO patients [10]. In atherosclerosis patients, anaemia is an independent risk factor for CVD outcomes [151]. In vasculitis, anaemia is a common condition that is more related to ANCA associated vasculitis (AAV) [152]. Therefore, it seems that the pathogenetic mechanisms of TAO and vasculitis can cause anaemia, while in atherosclerosis, anaemia is just an independent risk factor for CVD. These findings show that the underlying conditions unrelated to atherosclerosis are probably responsible for causing anaemia in atherosclerosis patients. Meanwhile, in TAO and vasculitis, anaemia should be considered in the context of these diseases as a part of them [9].

Different triggers, such as infection, inflammation pathways, and autoimmune responses, can induce elevated levels of CRP and ESR. These triggers usually are common in these three diseases, despite the origin, type, and pathway of them being specific in each disease. However, CRP and ESR do not have enough specificity for each pathway and are elevated in all of them, so we cannot recognize which one was responsible. Furthermore, some pain killers like NSAIDs (nonsteroidal anti-inflammatory drugs), usually taken by TAO patients, can influence the result of ESR [153].

In atherosclerosis, thrombosis is a consequence and product of the disease and happens suddenly as an event. Whilst in TAO and vasculitis, this is a part of pathogenesis and occurs consistently. As a marker that indicates the thrombosis and fibrinolysis situation, D-dimer has an elevated level in TAO and vasculitis consistently, while in atherosclerosis it rises during a thrombotic event [111].

### 9.3. Differential Biomarkers

Nitric oxide, MPV, TLRs, MDA, ox-LDL, sST2, antioxidant system, autoantibodies, and type of infection are differential markers.

Numerous studies show that although the diagnosis of TAO is ambiguous, despite the similarities, there are still significant differences between TAO, atherosclerosis, and vasculitis among laboratory markers that could be routinely effective in the differential diagnosis of TAO. Therefore, by examining these differences more closely, it is possible to design differential diagnostic panels and differentiate the TAO from atherosclerosis and vasculitis from the beginning of the patient’s visit to the laboratory with the simplest, routine, and, of course, cheapest tests.

In general, our studies show that, for laboratory differential diagnosis between TAO and atherosclerosis or vasculitis, tests, such as CBC, Cholesterol, HDL and LDL, FBS, ANA, Anti-dsDNA, and Lupus anticoagulant, should be considered first. Then, in more advanced stages after the initial tests, more detailed diagnostic charts can be designed. In these charts, Anti-Cardiolipin (IgM class), Anti-beta2 (IgM class), ANCA, oral bacteria and rickettsia, oxLDL, and other mediators such as TLR4 can be tested. Evaluating ESR, CRP, or protein C and S is not recommended as diagnostic biomarkers for TAO (Table 7 and Table 8).

## 10. Conclusions

Unfortunately, even after more than a century since Buerger first defined TAO, its diagnosis continues to be challenged. To date, no definitive biomarker points to a TAO diagnosis in patients with unusual manifestations of the disease or in patients with the risk factors of atherosclerosis or autoantibodies.

Until recently, the treatment of TAO has most often proceeded as a treatment for peripheral arterial disease rather than vasculitis. TAO patients usually undergo angiography to determine if they are eligible for angioplasty or bypass surgery, whilst the long-term outcomes of such procedures are unfavourable. At the same time, the outcomes of TAO patients who receive immunosuppressant medications, as in other types of vasculitis, are highly variable and typically unfavourable.

In this review, we noted both similarities and differences between TAO and atherosclerosis or vasculitis. These differences indicate that TAO is a disease entity distinct from peripheral arterial disease or vasculitis. In addition, the common biomarkers showed several common mechanisms in the development of TAO.

The tendency towards indistinctness regarding TAO may arise from the reductionism methodology favoured by researchers and the mechanistic thinking employed in the study of this disease. In addition, researchers tend to break down TAO into separate parts and topics for study without considering the aggregate interconnectedness of the disease as a whole. It is, after all, easy to lose sight of the whole when we focus our attention on individual parts. Achieving a holistic view of TAO instead of focusing on its dissimilarities to ASO or vasculitis can result in seeing the commonalities between TAO and other vascular diseases for a better understanding of this disease.

## Figures and Tables

**Table 1 diagnostics-11-01736-t001:** Summary of the main findings on Complete Blood Count (CBC).

CBC	TAO	Atherosclerosis	Vasculitis
Anaemia	Has been reported	Independent factor	Common
Platelet count	Mostly normal; rarely high in patients with major amputation or low due to consumption in thrombosis formation	Normal	Thrombocytopenia is common
Leukocyte count	Leukocytosis with neutrophilia has been reported	Mostly high and considered as a risk factor	Leukopenia is common
MPV	Mostly low	Mostly high due to hyperactivation	Mostly normal

**Table 2 diagnostics-11-01736-t002:** Summary of the main findings on biochemical markers.

Biochemical Markers	TAO	Atherosclerosis	Vasculitis
Fast Blood Sugar	Normal to slightly high (< 120 mg/dL)	Normal to high	Normal
Cholesterol	Normal to slightly high	High	Normal
LDL	Normal to slightly high	High	Normal
HDL	Low to normal	Low	Normal
Total oxidative stress	High with intact antioxidant capacity	High with impaired antioxidant capacity	High in several autoimmune vasculitis
MDA	Normal	High	High
NO	Low with eNOS polymorphism	High	Low with eNOS polymorphism High in SLE and RA with iNOS high expression
Creatinine	High	Normal to slightly high	Slightly high in Anca-Associated Vasculitis (AAV) and normal in Behcet’s disease
Heavy metals	High level of lead, cadmium, arsenic	High level of lead, cadmium	High level of lead and is related to mercury in Wegener’s granulomatosis
Catecholamine concentration	High level of adrenalin and noradrenalin	High level of epinephrine and norepinephrine	High especially in catecholamine-induced vasculitis
Bilirubin	Normal	Low to High High levels associated with good prognosis	High in AILD with small vessel vasculitisLow in Behcet’s disease
Hepatic function test (AST and ALT)	High AST with normal ALT	Mostly high level is related to fatty liver	High in liver involvement related to AAV and Rheumatoid Arthritis

**Table 3 diagnostics-11-01736-t003:** Summary of the main findings on inflammatory markers.

Inflammatory Biomarkers	TAO	Atherosclerosis	Vasculitis
ESR	Controversial	High (usually >18 and <30 mm/h)	Extremely high (even >100 mm/h)
CRP	Controversial	High (usually >12 mg/L)	High (usually 5 to 15 mg/L) More than 25 mg/L as a marker of infection in vasculitis
Pro-inflammatory	High level of TNF-α, IL-1,IL-6	High level of IL-1, IL-6, IL-18	High level of TNF-α, IL-1,IL-6
Th1-related	High level of IL-12 IFN-γ	High level of IL-12	High level of IFN-γ, IL-12,
Th2-related	High level of IL-4, IL-5, IL-13	High level of IL-4, IL-5	High level of IL-4, IL-5, IL-13
Th17-related	High level of IL-17, IL-22, IL-23, IL-21	High level of IL-21, IL-22, IL-23	High level of IL-23, IL-17, IL-21
Anti-inflammatory	Low level of IL-10	Low levels of TGF-β, IL-10	High level of IL-10
Others	High level of IL-33 (without increasing levels of Soluble suppression of tumorigenicity-2, sST2)	High level of IL-35, IL-33, IL-15	High level of IL-33 (with increased sST2 level)
HMGB1	High	High	High
Neopterin	High in the acute phase	High	High
TLRs	Low level of TLR4	High level of TLR4 and TLR2	High level of TLR4 and TLR5
Matrix Metalloproteinases (MMPs)	High MMP-9	High MMP-1, MMP-2, MMP-9, MMP-3, MMP-12, MMP-13	High MMP-2, MMP-9, MMP-3
ICAM1	High	High	High in AAV, SLE, WG
Complement component	High level of C4 in patients with severe form of TAO	High serum level of C3, C4	High C3, C4 levels in immune complex-mediated vasculitis are reported while in SLE, C3 and C4 were decreased

**Table 4 diagnostics-11-01736-t004:** Summary of the main findings on autoantibodies.

Autoantibodies	TAO	Atherosclerosis	Vasculitis
**Anti-Cardiolipin**	positive (IgM class)	Negative	positive (mostly IgG class)
**Anti-dsDNA, ANA**	Negative	Negative	Positive in vasculitis such as SLE
**Anti-beta2**	positive (IgM class)	positive (IgM class)	Not reported
**AECA**	positive	Negative	positive
**ANCA**	negative	Negative	Positive in AAV
**Lupus anticoagulant**	negative	Negative	positive

**Table 5 diagnostics-11-01736-t005:** Summary of the main findings on thrombotic factors.

Thrombotic Factors	TAO	Atherosclerosis	Vasculitis
Hypercoagulation activity	Mostly considered as normal, but with increased risk with Factor V _Leiden_ mutation	Normal activity	Hypercoagulable states mostly reported in Behçet syndrome and AAV
d-dimer	It has been reported D-dimer levels are considerably elevated in patients with TAO	High in thrombotic event	High levels in AAV, SLE, Granulomatosis with polyangiitis (GPA) Henoch-Schönlein purpura, Kawasaki disease
Fibrinogen level	High level	High	High in systemic vasculitis
Protein C, Protein s	Mostly considered as normal	Protein C and Protein S deficiency has been reported as a risk factor for aggravation of atherosclerosis	Normal
Hyperhomocysteinemia	Has been reported	Has been reported	Has been reported

**Table 6 diagnostics-11-01736-t006:** Summary of the main findings on infection.

	TAO	Atherosclerosis	Vasculitis
Infection	Oral bacteria and rickettsial infection.	Chlamydophila pneumoniae, Helicobacter pylori, CMV (Cytomegalovirus) and oral bacteria	Viral infection (e.g., Human Immunodeficiency Virus and CMV), Streptococcus pneumoniae, rickettsia, Mycobacterium, Staphylococcus aureus, Chlamydia, Neisseria

**Table 7 diagnostics-11-01736-t007:** Routine laboratory differential diagnostic tests for TAO and Atherosclerosis.

Disease	TAO	Atherosclerosis
Biomarker		
Cholesterol	Normal to slightly high	High
LDL	Normal to slightly high	High
HDL	Low to Normal	Low
FBS	Normal to slightly high	Normal to high (in diabetic patients)
Anti-Cardiolipin	Positive (IgM class)	Negative
Protc, prots	Normal	Prot c and prot s deficiency
MPV	mostly Low	mostly High
oxLDL	Normal	High
Oral bacteria and rickettsial infection.	Positive	Negative for rickettsial infection
TLR-2 and -4	Low level of TLR4	High level of TLR4 and TLR2

**Table 8 diagnostics-11-01736-t008:** Routine laboratory differential diagnostic tests for TAO and Vasculitis.

Disease	TAO	Vasculitis
Biomarker		
ESR	Controversial	High
CRP	Controversial	High
Anti-dsDNA, ANA	Negative	Positive
Lupus anticoagulant	Negative	Positive
Anti-Cardiolipin	Positive	Positive
Anti-beta2	Positive	Not reported
ANCA	Negative	Positive
Oral bacteria and rickettsial infection.	Positive	Positive for rickettsial infection
Hypercoagulation activity	Mostly considered as normal but with increased risk with Factor V _Leiden_ mutation	hypercoagulable states mostly reported in Behçet syndrome and AAV
Platelet count	Mostly normal; (rarely high or low in specific conditions)	thrombocytopenia
Leukocyte count	Leukocytosis with neutrophilia	Leukopenia
TLR	Low level of TLR4	High level of TLR4 and TLR5

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
