# Peer review of "Recent Updates and Advances in Winiwarter-Buerger Disease (Thromboangiitis Obliterans): Biomolecular Mechanisms, Diagnostics and Clinical Consequences"

_diagnostics, 2021, doi:10.3390/diagnostics11101736_

Round 1
Reviewer 1 Report
Fazeli and colleagues reviewed updates and advances in Winiwarter-Buerger Disease (Thromboangiitis Obliterans, TAO). They mainly describe a large number of controversial findings without emphasizing potential directions. Due to some limitations in English, the manuscript is relatively difficult to follow. Some examples below:
Line 54: except for smoking – please correct
Line 96: Other Metabolic Disorders – do not capitalize
Line 112: has been reported ranged – please correct
Line 156: “the backup of nitric oxide, so called heme-oxigenase 1 (HMOX1) is intact – is unclear. Please elaborate.
Line 168: “Also, increased and oxLDL are not different from smokers [35]” is unclear.
Line 210: “This aspect has been recently arise again” please correct.
Table 3: Alterations in MMPs should be specified.
Line 408: “Based on this approach, there are two groups of markers; (1) differential markers and 408 (2) similar markers, (3) controversial markers.” should be three groups of markers. Please discuss in the same order below.
Tables 7 and 8 are most relevant for clinical practice and should be clearly highlighted and discussed.
Author Response
We Would like to thank the Expert Reviewer 1 for the efforts to improve our ms: this is much appreciated.
We modified and improved our ms according to all the suggestions of the Reviewer.
Prof. F Mannello (on behalf of all Authors)
Reviewer 2 Report
In this narrative review, the authors reported in detail disease biomarkers potentially useful to differentiate Thromboangiitis Obliterans (TAO) from atherosclerosis and vasculitis. More in detail, the authors focused on relevant biomolecular and laboratory para-clinical markers, including pro-inflammatory and cell-mediated immunity cytokines as well as autoantibodies. Results of the review were that there are no specific biomarkers of TAO.
I found the study rather interesting since the authors focused on a relevant but underrecognized vasculitis potentially causing severe disability. Although the review is well-organized and each session is balanced, I have some concerns about the study that should be addressed by the authors.
- It is unclear why the authors compared TAO only with two separate types of diseases affecting the circulatory system: atherosclerosis and vasculitis. Also, it is unclear what types of vasculitis are included under the group of “vasculitis”.
- Besides biomolecular and laboratory biomarkers of TAO, the authors should expand the discussion by including relevant pathological as well as instrumental findings in TAO. More in detail, the authors should report relevant pathological biomarkers of vasculitis and also relevant instrumental exams useful in TAO, including magnetic resonance or echography.
- The authors should add further relevant data about the clinical picture of TAO. Specifically, they should report clinical findings useful for the diagnosis of TAO and other vasculitides. Also, the authors have no reported any study about the involvement of the central nervous system in TAO and vasculitis.
- The review lacks relevant information about clinical scales useful for the clinical examination of patients with TAO and other vasculitides.
- The authors should report data about genetic testing in TAO.
- Relevant therapeutic interventions in TAO and other vasculitides should be added to the discussion.
- The conclusion should include perspectives about public health strategies to face the TAO in the future.
Author Response
We would like to thank the Expert Reviewer 2 for the constructive and insightful suggestions improving our ms: this is much appreciated.
We have modified the ms according to all the comments of the Reviewer.
All the best
Prof. F Mannello (on behalf of all Authours)

Round 2
Reviewer 1 Report
The authors performed the necessary corrections. It would have been easier to follow if they would have used a track changes option and specified the modifications in the response. Anyway, I have no additional suggestions. Spelling and grammar should be checked again during the publication process.
Reviewer 2 Report
The authors have addressed my concerns. I have no further comments. I only suggest to the authors to revise the manuscript by editing english language.